# Investigation of Impulse and Continuous Discharge Characteristics of Large-Capacity Lithium-Ion Batteries

**Sergey V. Kuchak \* and Sergey V. Brovanov**

Department of Electronics and Electrical Engineering, Novosibirsk State Technical University, Novosibirsk 630073, Russia
\* Correspondence: kuchak.2012@corp.nstu.ru

**Abstract:** Lithium-ion batteries are one of the most popular and efficient energy storage devices. In this paper, the characteristics of high-capacity lithium-iron-phosphate batteries during the impulse and long-term operation modes of batteries with different levels of the discharge current are considered. A modified DP-model is proposed. The novelty of the model is the possibility to calculate the activation polarization parameters for different discharge currents. The state of charge is estimated using a high-order polynomial. Based on the developed model, transient processes with rapid load changes and the dependence of the battery voltage on the state of charge were obtained. Here, the model is intended to be used for the design of energy storage systems. The results showed that the DP-model is reliable under the tested conditions and can be used for the considered application.

**Keywords:** lithium-ion battery; equivalent circuit; discharge characteristics; simulation model

## 1. Introduction

Renewable energy sources and electric vehicles in recent years have experienced a tremendous growth in the application of energy storage systems (ESS).

The use of specialized control algorithms leads to power system efficiency increases both at the level of consumers and generation [1–3]. ESS allows to store electrical energy during low-consumption periods and to generate at a shortage of generation capacity. In addition, the ESS uses allow obtaining an improvement of the stability and uninterrupted power supply.

Lithium-ion batteries (*LIB*) remain one of the most promising electrical energy storage devices. Main advantages of the *LIB* are relatively large gravimetric and volumetric density, low self-discharge rate, long life, no memory effect [4–6], and low material toxicity class [7]. Despite their popularity, *LIB*s remain the object of research and need new modeling methods to realize an accurate whole-life-cycle prediction by effectively considering the current, voltage, and temperature variations.

There are various approaches to the formation of *LIB* models. Electrochemical models [8,9] are a description of the physical and chemical processes occurring in the battery. The result is a system of nonlinear differential equations. This approach allows achieving high accuracy of the model. However, it is complex in terms of the formation and application of the model.

Equivalent circuits [10,11] allow forming easy-to-understand models based on the basic elements of electrical engineering: resistors, capacitors, inductances, and current or voltage sources. There are various battery equivalent circuits: from the R-model (EMF-source and resistance) to DP-models (series connection of two RC-circuits, resistance, and EMF-source).

Statistical models [12,13] are created on the basis of a set of experimental data in the required modes. Further processing of the results can be carried out using mathematical expressions or neural algorithms. This method is easy to implement and apply. However, it

is not characterized by high accuracy, because it does not take into account a large number of factors that affect battery performance.

This paper discusses the combined application of two methods: statistical and equivalent circuits. This approach was developed in [14–16]. Using the first method, the dependence of the open circuit voltage (*OCV*) on the state of charge (*SoC*) of the battery (continuous processes) is described. The second method considers the processes of an impulse discharge (short-term transitions). For example, in [17], an improved feedforward-long short-term memory (FF-LSTM) modeling method is proposed to realize an accurate whole-life-cycle *SoC* prediction by effectively considering the current, voltage, and temperature variations. An optimized sliding balance window is constructed for the measured current filtering to establish a new three-dimensional vector as the input matrix for the filtered current and voltage.

Furthermore, in [18], a characterization method for a lithium-iron-phosphate (LFP) pouch cell is presented and evaluated, using a method that applies to hybrid current pulses called hybrid power pulse characterization (HPPC). The model is intended to be used for the development of electrical mobile applications, such as electric vehicles (EV) and electric vehicle supply equipment (EVSE), where high capacity and currents are required through the cell.

The disadvantage of most studies is the study of low-capacity batteries: fractions and units of ampere-hours. For most applications in the energy sector, such as high-power energy storage systems, operating DC voltage system, current values reach tens and hundreds of amperes. Thus, the reliable model creation of a high-capacity lithium-ion battery is an actual task.

## 2. Materials and Methods

The DP-model (dual polarization) was selected as base. The equivalent circuit is shown in Figure 1. The model is an electrical equivalent circuit, where $U_{OCV}$ is the open circuit voltage; $R_{int}$ is the internal ohmic resistance to direct current; $C_{PA}$, $R_{PA}$ are the capacitance and resistance of the activation polarization, respectively; $C_{PC}$, $R_{PC}$ are the capacitance and resistance of the concentration polarization, respectively; $U_T$ is the voltage at the terminals of the element.

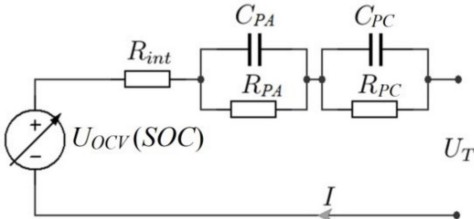

**Figure 1.** DP-equivalent circuit of battery.

The parameters of the concentration polarization reflect the processes of ion concentration changing in the near-electrode layer of the electrolyte and have a significant effect on the impulse processes. The activation polarization parameters correspond to the processes on the electrode, in this case, the intercalation of lithium ions, and are relevant in longer processes [19].

If the mutual influence of the circuits is not taken into account, the expression for the battery terminal voltage can be presented as follows:

$$U_T(t) = U_{OCV}(SoC) - \Delta U_{INT} - \Delta U_{PA} \cdot (1 - e^{-\frac{t}{\tau_{PA}}}) - \Delta U_{PC} \cdot (1 - e^{-\frac{t}{\tau_{PC}}}) \quad (1)$$

where $U_{OCV}(SoC)$ is the voltage before the current impulse (*OCV*); $\Delta U_{INT} = I \cdot R_{int}$ is voltage drop across the DC ohmic resistance; $\Delta U_{PA} = I \cdot R_{PA}$ and $\Delta U_{PC} = I \cdot R_{PC}$ are voltage drops on the activation and concentration polarization circuits, respectively; $\tau_{PA} = R_{PA} \cdot C_{PA}$

and $\tau_{PC} = R_{PC} \cdot C_{PC}$ are the time constants of the activation and concentration polarization circuits, respectively.

LT-LYP380AH are the batteries under study. Their rated voltage is 3.2 V and the rated capacity is 380 Ah. The cathode material is lithium iron phosphate $LiFePO_4$, and carbon is used as the anode material. The electrolyte is a solution of lithium salts $LiPF_6$ in a mixture of organic solvents.

To obtain the parameters of the equivalent circuit, a comprehensive study was carried out. As a result, the following characteristics were obtained:

- Time charts of *LIB* voltage at different impulse discharge currents;
- Dependence of open circuit voltage on the state of charge;
- Discharge characteristics at different discharge currents [20].

Based on the characteristics of impulse discharges, the values of passive elements (resistances and capacitances) of the DP-model were obtained. The results of this study are discussed in Section 3.1 to describe long-term processes; the study of the dependence of the open circuit voltage on the depth of discharge is carried out. The final stage of the study is the simulation of discharge characteristics. The study of long-term processes is discussed in Section 3.2. It should be noted that the methods used are suitable for other battery chemistries. Probably, the differences will be in the values and features of the parameters of the equivalent circuit. Moreover, the discharge characteristics will be different.

## 3. Results

### 3.1. Impulse Discharges

For these tests, 3 batteries were selected. The selected batteries had a spread in discharge capacity of less than 0.05 C. Before discharge, the batteries were charged by a two-stage method, then kept in a currentless state for 30 min. The voltage value before the impulse was 3.36 V. The tests were carried out under normal climatic conditions. During the experiment, current impulses with amplitudes of 0.2 C, 1.0 C, 1.5 C, and 2.5 C, respectively, were applied to the *LIB*. The impulse duration was 8 s.

Figure 2 shows the resulting voltage charts for impulse discharge of *LIB*. The curves are averaged data for the tested batteries.

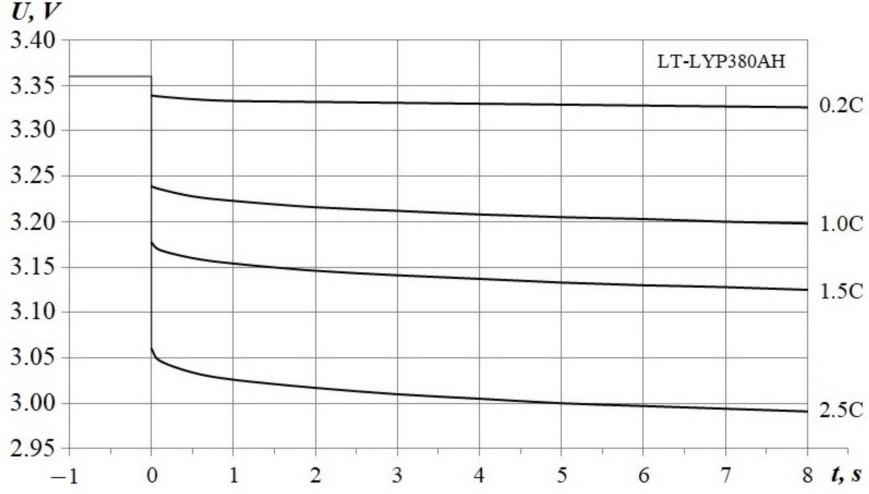

**Figure 2.** Voltage at impulse discharge.

### 3.1.1. Determination of Circuit Parameters

As can be seen from the charts, a significant part of the voltage drop occurs instantly at the beginning of the transient process. This reflects the active nature of the voltage drop. Further, an almost exponential voltage decrease occurs. Based on the voltage shape, it is possible to determine the value of the resistance $R_{int}$ as the ratio of the difference in

the voltage drop at the impulse beginning for different currents to the difference of these currents:

$$R_{int} = \frac{\Delta U_{2.5C} - \Delta U_{0.2C}}{2.5C - 0.2C} = \frac{U_{0.2C}(0+) - U_{2.5C}(0+)}{2.5C - 0.2C} = \frac{3.338 - 3.06}{950 - 76} = 318.1 \text{ μOhm} \quad (2)$$

where $\Delta U_{2.5C} = U(0\text{-}) - U_{2.5C}(0+)$ and $\Delta U_{0.2C} = U(0\text{-}) - U_{0.2C}(0+)$ are voltage drops for 2.5 C- and 0.2 C-discharge currents, respectively, $U(0\text{-})$ is voltage value before current change, $U_{2.5C}(0+)$ and $U_{0.2C}(0+)$ are voltage values after current change for 2.5 C- and 0.2 C-discharge currents, respectively.

Moreover, from Figure 2, it is possible to determine the $U_{OCV}$ and $\Delta U_{INT}$ values, and also the $U(t)$ values at times $t$ = 0.5, 1.0, 4.0, 8.0 s. Based on these values, the system of equations was solved for $\tau_{PC}$, $U_{PA}$, and $U_{PC}$. It was determined that the time constant of concentration polarization processes $\tau_{PC}$ weakly depends on the current value and averages 0.294 s. At the same time, taking into account the magnitude of the voltage drop, it was obtained that the resistance $R_{PC}$ has a value of 26.14 μOhm $\pm$ 6% for modes 1.0 C, 1.5 C, and 2.5 C. Based on the obtained $R_{PC}$ value and the time constant $\tau_{PC}$, the $C_{PC}$ capacity is determined as:

$$C_{PC} = \frac{\tau_{PC}}{R_{PC}} = \frac{0.294}{26.14 \cdot 10^{-6}} = 11.25 \text{ kF} \quad (3)$$

In a similar way, the value of the time constant of the activation polarization circuit $\tau_{PA}$ was obtained. It was determined that this value is practically independent of the current and its value was 5.35 s $\pm$ 6.5%. In this case, the spread in the value of the resistance $R_{PA}$ and capacitance $C_{PA}$ was $\pm$29%. Figure 3 shows the obtained dependences of these parameters on the discharge currents. In the charts, the calculated values of $R_{PA}$ and $C_{PA}$ for the corresponding currents are shown as markers. The solid lines represent the graphs of the approximation of the results.

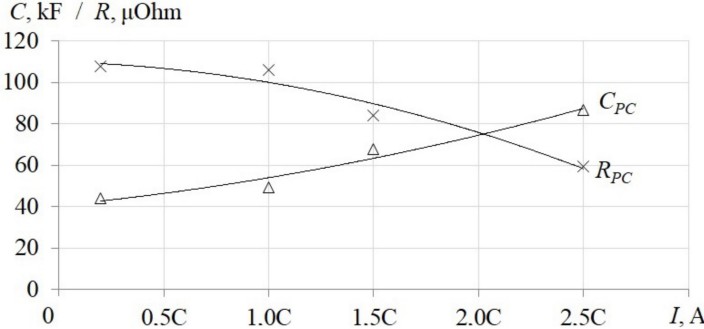

**Figure 3.** Parameters of the activation polarization equivalent circuit for different discharge currents.

### 3.1.2. Circuit Simulation

Confirmation of the compliance of the obtained parameters of the equivalent circuit for impulse discharges was carried out using the simulation of electrical processes in *PSIM*. The generated model is shown in Figure 4. Here, the source voltage $U_{OCV}$ is set to 3.36 V. This corresponds to a fully charged battery state. Figure 5 shows a comparison of experimental data (markers) and simulation results (solid lines) for 0.2 C, 1.0 C, 1.5 C, and 2.5 C discharge currents, respectively.

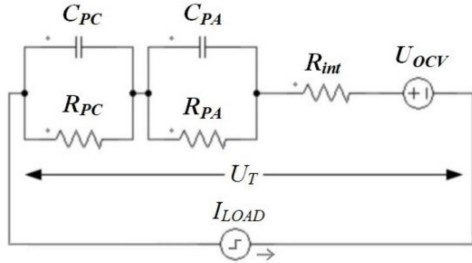

**Figure 4.** Battery simulation model.

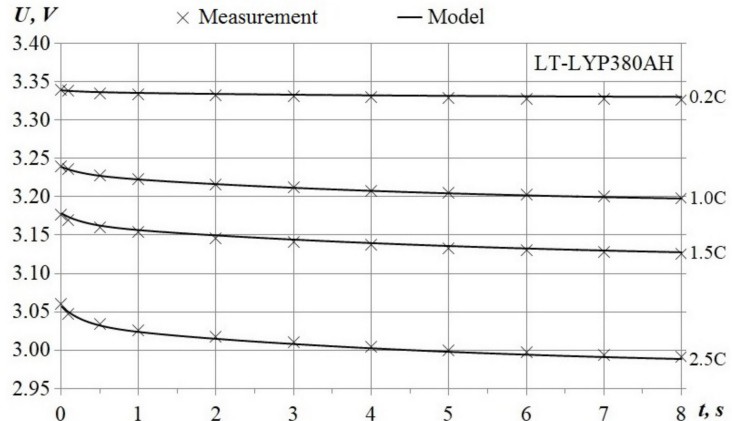

**Figure 5.** Comparison of measurement and simulation impulse discharges.

The results of *LIB* modeling correspond with the minimum error to the results of the experiment, which confirms the correctness of the calculations. The average and maximum errors in this model were 0.06 and 1.14%, respectively. The disadvantage of this implementation is the non-universality of the model, since it is necessary to preliminarily set the corresponding values of the parameters of the concentration polarization circuit $R_{PA}$ and $C_{PA}$ for each current. In addition, the quality of the transient process is influenced by the mutual influence of RC-circuits.

### 3.1.3. Simulation of the Model with Current Feedback

To obtain a model with automatic setting of the $R_{PA}$ and $C_{PA}$ parameters, the dependences of these parameters on the discharge current (see Figure 3) were averaged by second-order polynomials. As a result, the following expressions are obtained:

$$R_{PA}(I) = A_R \cdot I^2 + B_R \cdot I + C_R \tag{4}$$

$$C_{PA}(I) = A_C \cdot I^2 + B_C \cdot I + C_C \tag{5}$$

where $A_R = -49.5 \cdot 10^{-12}$, $B_R = -7.17 \cdot 10^{-9}$, $C_R = 110 \cdot 10^{-6}$, $A_C = 24.7 \cdot 10^{-3}$, $B_C = 25.8$, $C_C = 40.6 \cdot 10^3$.

The result of these expressions corresponds to the calculated values with less than 10% error, which indicates sufficient accuracy in setting the characteristics. The proposed mathematical model of *LIB* with current feedback (*CFB*) is shown in Figure 6.

In the model, the battery is presented as a regulated voltage source $U_T$, the control system of which can be described by expressions (1), (4), (5). Block $T_1$ is a resettable countdown timer for generating a time reference for the exponent. The timer is reset when the *LIB* current changes rapidly.

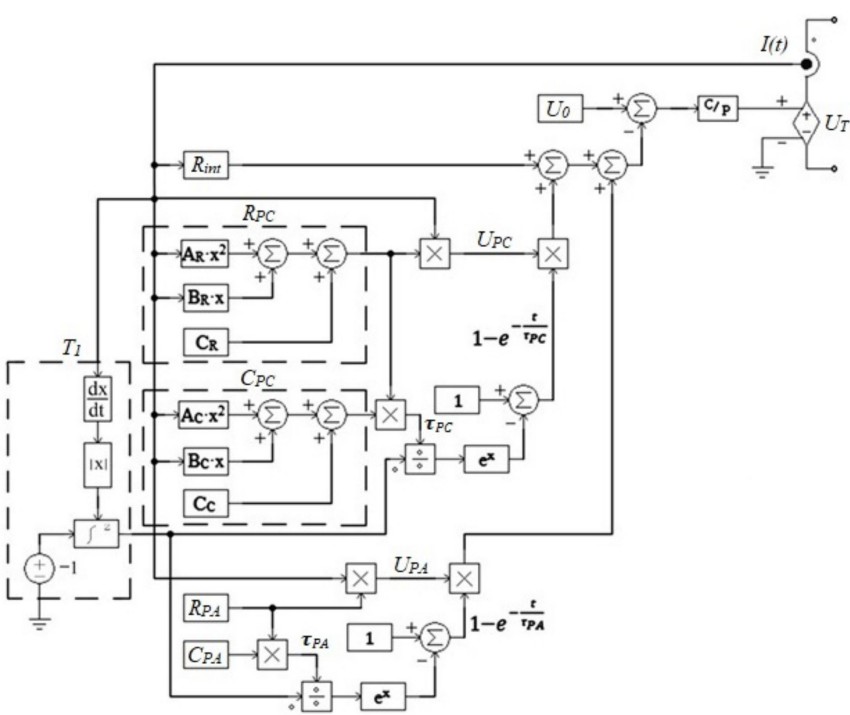

**Figure 6.** Simulation model of *LIB* with current feedback.

The time charts obtained by modeling this circuit, combined with the experimental charts, are shown in Figure 7. It can be seen from the charts that the introduction of the $R_{PA}$ and $C_{PA}$ dependences on the current into the system did not lead to deterioration in the numerical indicators reflecting the transient process of the *LIB* impulse discharge. The average and maximum errors in this model were 0.05 and 0.12%, respectively.

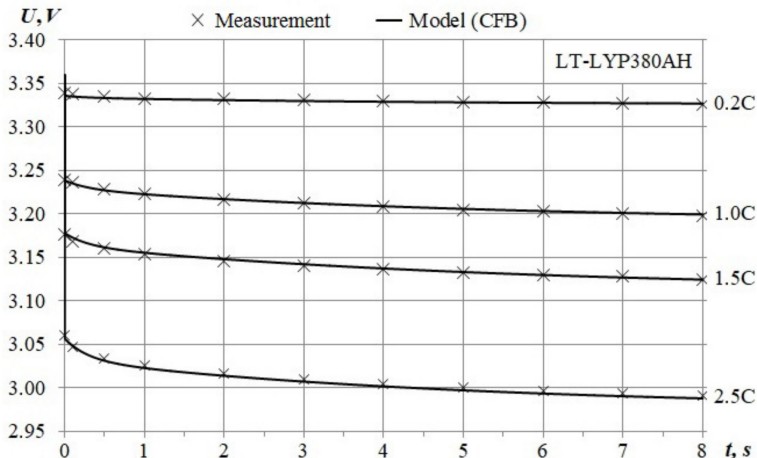

**Figure 7.** Comparison of measurement and simulation with current feedback.

### 3.2. Open Circuit Voltage Dependence on Depth of Discharge

The next step is to introduce the dependence of the *OCV* on the state of charge. The discharge characteristic of the *LIB* at low current discharge *LCD* (5 A, 0.013 C) was taken as the basis for modeling. For the measurements, four *LT-LYP380AH* batteries were selected. The batteries were preliminarily charged by a two-stage method, after which they were kept for 30 min in a currentless state. Further, the discharge was carried out until a voltage of 2.5 V was reached. Figure 8 shows the averaged result for the tested batteries.

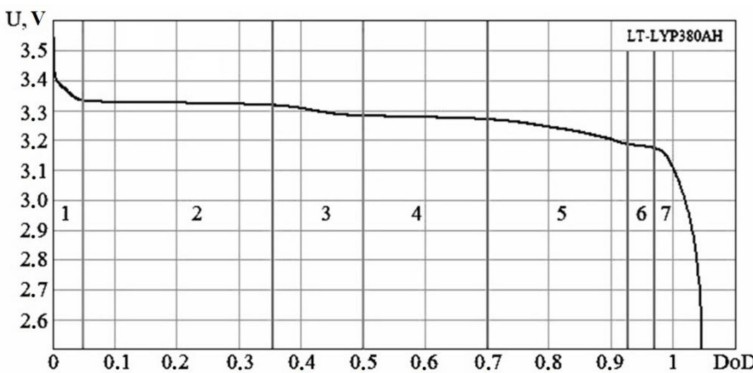

**Figure 8.** The *LIB* voltage experimental curve for different depths of discharge.

The charts show the dependence of the voltage on the battery on the depth of discharge (*DoD*), determined by the following expression:

$$DoD = 1 - SoC \tag{6}$$

where *SoC* is the state of charge defined as:

$$SoC = \frac{C_{CHRG}}{C_{NOM}} \tag{7}$$

Here, $C_{CHRG}$ (Ah) is the charge received, $C_{NOM}$ (Ah) is the nominal capacity of the battery.

The dependence has a pronounced nonlinear character. The chart can be divided into 7 sections, which are described in Table 1. According to the table, the battery *SoC* can be determined from the voltage value at the battery terminals with an accuracy of ±15%, which is sufficient for most applications.

**Table 1.** Description of the *LIB* voltage on the depth of discharge curve.

| № | *DoD* Range | Separate Sections | Voltage Range, V |
|---|---|---|---|
| 1 | 0–0.05 | Sharp decline | 3.55–3.33 |
| 2 | 0.05–0.35 | Linear, horizontal | 3.33–3.32 |
| 3 | 0.35–0.50 | Smooth decline | 3.32–3.28 |
| 4 | 0.50–0.70 | Linear, horizontal | 3.28–3.27 |
| 5 | 0.70–0.93 | Smooth decline | 3.27–3.18 |
| 6 | 0.93–0.97 | Linear, horizontal | 3.18–3.17 |
| 7 | 0.97–1.05 | Sharp decline | 3.17–2.50 |

It should be noted that the current magnitude is small. Based on this, the influence of polarization processes is negligible. This statement was confirmed by a series of experiments. In the experiments, the *OCV* was measured in pauses during interval charge and discharge (intervals of 0.2 $C_{NOM}$ and the last one until voltage decreases to 2.5 V). A comparison of the characteristics is shown in Figure 9. Dependency analysis showed that the relative error between *LCD* and *OCV* modes at the reference points (0.2, 0.4, 0.6, 0.8) is less than 1.4%. It should be noted that there is a negative characteristic deviation. For example, the average error in this range is minus 0.023 V (0.73% of the average voltage).

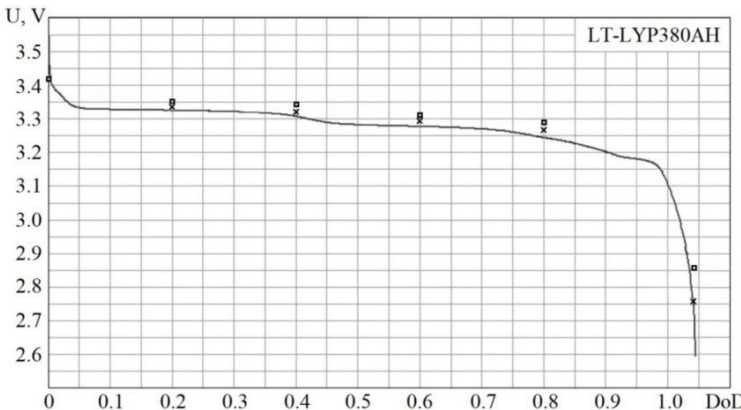

**Figure 9.** The *LIB* voltage curves for different depths of discharge. □—*OCV* at charging; ×—*OCV* at discharging; line—battery voltage at *LCD*.

### 3.2.1. Approximation of the Open Circuit Voltage Dependence

Since the curve has 7 characteristic sections, it can be represented as a polynomial of at least 8th order. The interpolation result corresponds to the following expression:

$$U_{OCV}(DoD) = \sum_{i=0}^{9} k_i \cdot DoD^i \tag{8}$$

where $k_i$ are the coefficients of the resulting polynomial (given the characteristic points of the curve, coefficients can be obtained by the "*polycoeff*" *MathCAD* Function).

The comparison of breakpoints of real characteristics and the polynomial curve is shown in Figure 10. The approximation was carried out according to the criterion of minimizing the error at the breakpoints and the absence of sections with increasing voltage. As can be seen from the figure, the graph corresponds to the main areas in the range from 0 to 97% of the depth of battery discharge. An analysis of the results at the break points showed that the maximum relative error in this range is no more than 0.58% of the measured values, which reflects a high degree of results matching. The relative error at the full discharge state is 9.22%.

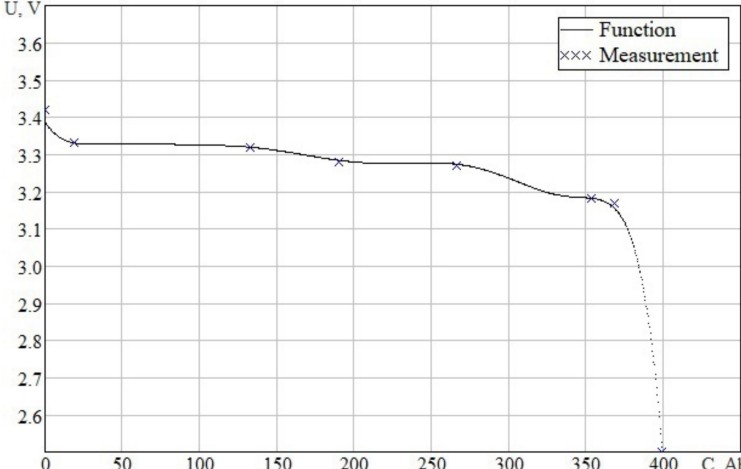

**Figure 10.** Comparison of polynomial curve and measured breakpoints for dependence of the *LCD* voltage on the discharged capacity.

As noted earlier, the curve has a negative deviation from the real dependence of the *OCV* on the depth of discharge. Therefore, a new error value was obtained: it is equal to minus 0.016 V (0.51%). This value is added to the coefficient $k_0$ of the polynomial.

Comparison of the modified polynomial curve with the *OCV* dependences on the discharged capacity is shown in Figure 11. Characteristics analysis showed that the relative error between the obtained characteristic and the reference points of the *OCV* dependences (0, 0.2, 0.4, 0.6) is less than 0.51%. The maximum error values at the last two breakpoints (0.8 and full discharge state) are 1.02 and 3.46%, respectively.

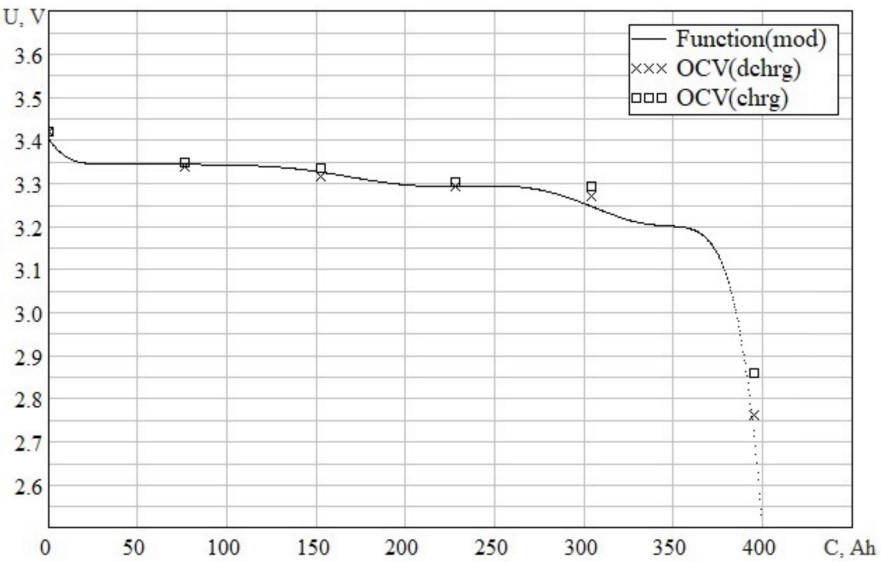

**Figure 11.** Comparison of modified polynomial curve and *OCV* dependences on the capacity.

### 3.2.2. Simulation of the Open Circuit Voltage Dependence

The *OCV* dependency implementation in *PSIM* is shown in Figure 12. The input signal *I(t)* is the current sensor feedback. The integrator *INT* converts the current value into the charged/discharged capacity. The time constant of the integrator is $-3600$ s. If this block is added to the model shown in Figure 6, it replaces the voltage reference block $U_0$.

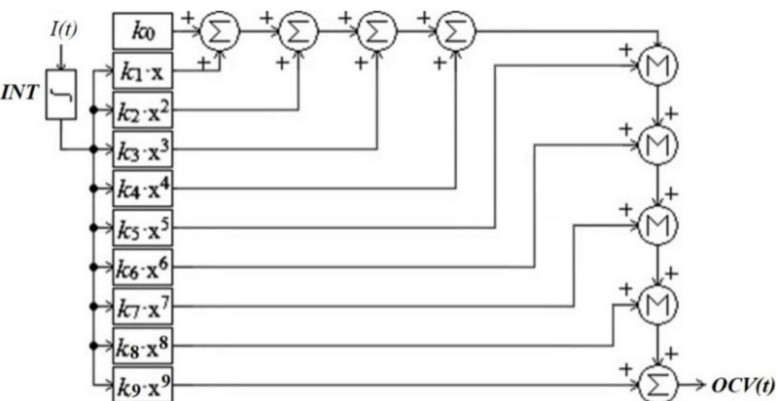

**Figure 12.** Implementation of the *OCV* dependence in *PSIM*.

Figures 13 and 14 show the experimental and simulated discharge characteristics, respectively. The characteristics follow the shape of the *OCV*. The analysis shows the following differences between the characteristics:

− Maximum average voltage error for the 375 Ah range corresponds to the 0.2 C chart, the error average value is less than 1.7% of the nominal voltage;
− An 8% difference between maximal discharge capacity for the 2.5 C-chart;
− The end-state-of-charge spread is less by 3.5% of the nominal capacity.

Deviations from the experimental characteristics can be explained by the fact that the model does not take into account thermal heat losses.

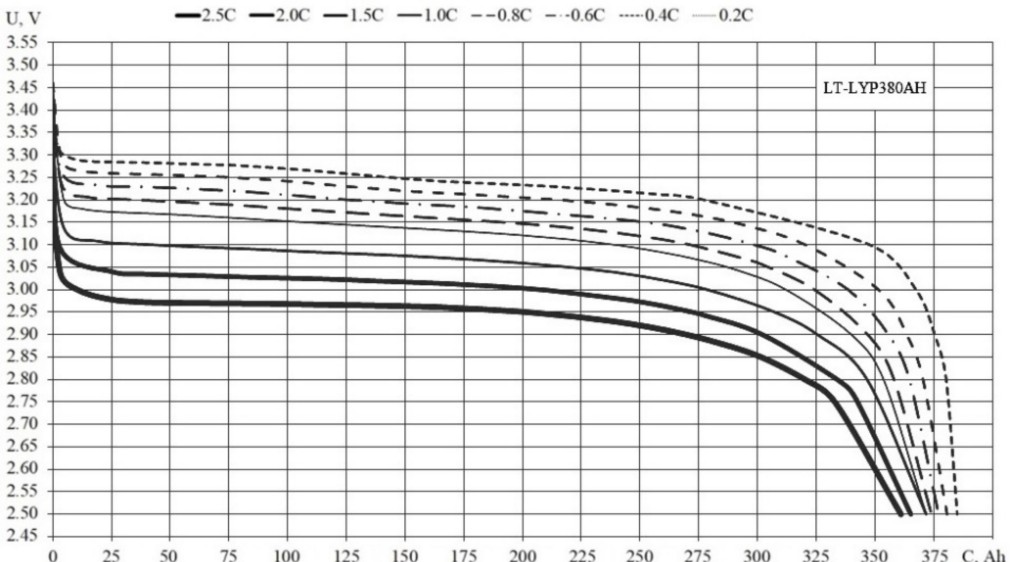

**Figure 13.** Experimental discharge characteristics.

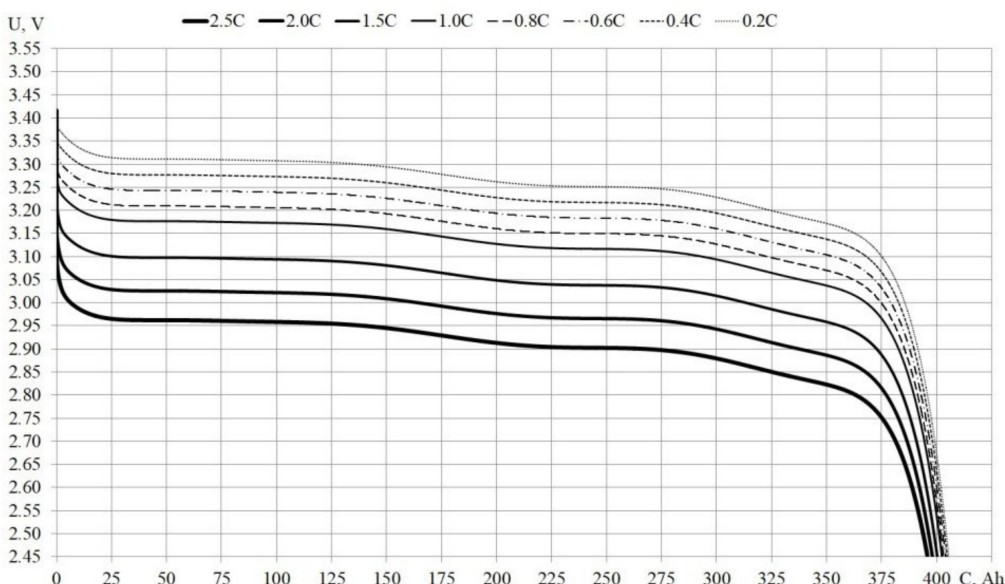

**Figure 14.** Simulated discharge characteristics.

### 4. Conclusions

A model of a high-capacity LiFePO$_4$ battery is proposed. The model is represented by a mathematical description of the actual characteristics of a 380 Ah battery. The DP model is taken as a basis. The dependence of the activation polarization parameters on the discharge current is presented. The state of charge is estimated by the 9th-order polynomial of voltage. The model was obtained by analyzing the impulse and continuous operation modes of the battery. As a result, a model was obtained that describes the impulse operation with an error of less than 0.2% and the dependence of voltage on the state of charge with an error of less than 2%. Discharge characteristics simulation showed acceptable results, which can be caused by the fact that heat losses are not taken into account. The model can be improved by taking into account the dependence of internal resistance on the state of

charge and battery temperature. The model can be useful for developers of high-power energy storage systems.

**Author Contributions:** Conceptualization, S.V.K. and S.V.B.; methodology, S.V.K.; software, S.V.K.; validation, S.V.B.; formal analysis, S.V.K. and S.V.B.; investigation, S.V.K.; writing—original draft preparation, S.V.K.; writing—review and editing, S.V.K. and S.V.B.; visualization, S.V.K.; supervision, S.V.B. All authors have read and agreed to the published version of the manuscript.

**Funding:** This research received no external funding.

**Data Availability Statement:** Not applicable.

**Acknowledgments:** The authors are grateful to Liotech LLC for the opportunity to conduct an experimental study of lithium-ion batteries.

**Conflicts of Interest:** The authors declare no conflict of interest.

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
