# Peer review of "Investigation of Impulse and Continuous Discharge Characteristics of Large-Capacity Lithium-Ion Batteries"

_processes, doi:10.3390/pr10122473_

Round 1

Reviewer 1 Report

The manuscript titled “Investigation of impulse and continuous discharge characteristics of large-capacity lithium-ion battery” can be accepted for publication after the following minor revisions.

1. Perform English editing of the manuscript. There are several grammatical errors and typos in the manuscript.

2. Emphasize the importance of Li-ion batteries compared to other technologies in the introduction section.

3. References in the manuscript (only 14?) are not enough to support the discussion and literature. Add additional references in appropriate places and discuss them properly.

Author Response

Response to Reviewer 1 Comments

Point 1: Perform English editing of the manuscript. There are several grammatical errors and typos in the manuscript. 

Response 1: Several grammatical errors and typos fixed

Point 2. Emphasize the importance of Li-ion batteries compared to other technologies in the introduction section.

Response 2: The main advantages of LIB are presented.

Point 3. References in the manuscript (only 14?) are not enough to support the discussion and literature. Add additional references in appropriate places and discuss them properly.

Response 3: 6 new references are added and reviewed.

Reviewer 2 Report

The authors discuss pulsed and long-term operation modes of batteries at different levels of the discharge currents. Based on the characteristics obtained, the model was developed that reflects transient processes during abrupt load changes and the battery voltage dependence on the state of charge.  The manuscript presents some significant results worthy of publication but should be revised to be more precise and accurate in the wording of the text. Below are some detailed comments:

1. The novelty of the work should be explained in detail in the Abstract, Highlights, and Conclusion. The final conclusion should be expressed with detailed experimental results.

2. The expression of the abstract should be improved. A more detailed presentation of innovation should be conducted. 

3. Please highlight the part improved by your own research and try to make it more obvious. 

4. More recent literature should be cited and analyzed for the mathematical analysis, such as An improved feedforward-long short-term memory modeling method for the whole-life-cycle state of charge prediction of lithium-ion batteries considering current-voltage-temperature variation, A Critical Review of Improved Deep Convolutional Neural Network for Multi-Timescale State Prediction of Lithium-Ion Batteries.

5. The writing of the paper should be improved. There are some grammar errors in the manuscript. It would be better to check the content in detail to ensure all expressions are correct. Please try to improve all the contents, the presentation should be revised to make the readers get your idea more clearly, making the logic clear and correct. Also, the innovation and realization should be clearly described.  

6. All the equations should be cited by number. The expression should be paid more attention to. The font size should be smaller than the main content. Please carefully check the sentences to make the expression to be correct.

I hope the authors consider my suggestion and make modifications or explanations for my review.

Author Response

Response to Reviewer 2 Comments

Point 1: The novelty of the work should be explained in detail in the Abstract, Highlights, and Conclusion. The final conclusion should be expressed with detailed experimental results.

Point 2: The expression of the abstract should be improved. A more detailed presentation of innovation should be conducted. 

Point 3. Please highlight the part improved by your own research and try to make it more obvious

Response 1 - 3: Abstract, Materials and Methods, Conclusions are modified in terms of novelty and features of the study.

Point 4. More recent literature should be cited and analyzed for the mathematical analysis, such as An improved feedforward-long short-term memory modeling method for the whole-life-cycle state of charge prediction of lithium-ion batteries considering current-voltage-temperature variation, A Critical Review of Improved Deep Convolutional Neural Network for Multi-Timescale State Prediction of Lithium-Ion Batteries.

Response 4. 6 new references are added and reviewed, including 2 of them on the latest LIB research.

Point 5. The writing of the paper should be improved. There are some grammar errors in the manuscript. It would be better to check the content in detail to ensure all expressions are correct. Please try to improve all the contents, the presentation should be revised to make the readers get your idea more clearly, making the logic clear and correct. Also, the innovation and realization should be clearly described.  

Response 5. The manuscript has been edited. Fixed several grammatical errors and typos.

Point 6. All the equations should be cited by number. The expression should be paid more attention to. The font size should be smaller than the main content. Please carefully check the sentences to make the expression to be correct. 

Response 6. Numbering errors fixed. Revised some mathematical expressions.

Reviewer 3 Report

This paper demonstrates a modelling steps of a specific battery model - LT-LYP380AH. Although the paper is easy to follow all steps taken in the modelling process, the first suggestion is to include a section called "Methodology", explaining the stages. It would help the readers to replicate the methodology to other models of batteries. 

Would this methodology be appropriatte for other battery chemistries? 

Take a look on the following references and try to incorporate some ideas in your text:

Nacu, R.C.; Fodorean, D. Lithium-Ion Cell Characterization, Using Hybrid Current Pulses, for Subsequent Battery Simulation in Mobility Applications. Processes 2022, 10, 2108. https://doi.org/10.3390/pr10102108

Please, correct the following mistakes:

line 78 - LT-LYP380AH is the battery under study. Its rated voltage is 3.2 V... (correct the sentence)

line 167 - Figure 7 (figure 6 is duplicated) 

Author Response

Response to Reviewer 3 Comments

Point 1: Although the paper is easy to follow all steps taken in the modelling process, the first suggestion is to include a section called "Methodology", explaining the stages. It would help the readers to replicate the methodology to other models of batteries.

Response 1: Several clarifications have been added.

Point 2: Would this methodology be appropriatte for other battery chemistries?

Response 2: Appropriate comments added to section 2.

Point 3: Take a look on the following references and try to incorporate some ideas in your text:

Nacu, R.C.; Fodorean, D. Lithium-Ion Cell Characterization, Using Hybrid Current Pulses, for Subsequent Battery Simulation in Mobility Applications. Processes 2022, 10, 2108. https://doi.org/10.3390/pr10102108

Response 3: The submitted article has been reviewed in  Introduction

Point 4: Please, correct the following mistakes:

line 78 - LT-LYP380AH is the battery under study. Its rated voltage is 3.2 V... (correct the sentence)

line 167 - Figure 7 (figure 6 is duplicated) 

Response 4: Fixed several grammatical errors and typos

Round 2

Reviewer 2 Report

The research content is well described for the revised version, including its innovation, logic, and experimental verification. It is suggested to be published.